# Different Seasonal Collections of *Ficus carica* L. Leaves Diversely Modulate Lipid Metabolism and Adipogenesis in 3T3-L1 Adipocytes

**DOI:** 10.3390/nu14142833

**Published:** 2022-07-10

**Authors:** Mariachiara Pucci, Manuela Mandrone, Ilaria Chiocchio, Eileen Mac Sweeney, Emanuela Tirelli, Daniela Uberti, Maurizio Memo, Ferruccio Poli, Andrea Mastinu, Giulia Abate

**Affiliations:** 1Department of Molecular and Translational Medicine, Division of Pharmacology, University of Brescia, 25123 Brescia, Italy; m.pucci003@unibs.it (M.P.); e.macsweeney@studenti.unibs.it (E.M.S.); e.tirelli004@unibs.it (E.T.); daniela.uberti@unibs.it (D.U.); maurizio.memo@unibs.it (M.M.); giulia.abate@unibs.it (G.A.); 2Department of Pharmacy and Biotechnology (FaBiT), University of Bologna, Via Irnerio 42, 40126 Bologna, Italy; manuela.mandrone2@unibo.it (M.M.); ilaria.chiocchio2@unibo.it (I.C.); ferruccio.poli@unibo.it (F.P.)

**Keywords:** seasonality, *Ficus carica* L., furanocoumarins, ^1^H-NMR profile, lipid metabolism, adipogenesis

## Abstract

Due to the high prevalence of obesity and type 2 diabetes, adipogenesis dysfunction and metabolic disorders are common features in the elderly population. Thus, the identification of novel compounds with anti-adipogenic and lipolytic effects is highly desirable to reduce diabetes complications. Plants represent an important source of bioactive compounds. To date, the antidiabetic potential of several traditional plants has been reported, among which *Ficus carica* L. is one of the most promising. Considering that plant metabolome changes in response to a number of factors including seasonality, the aim of this study was to evaluate whether *Ficus carica* leaves extracts collected in autumn (FCa) and spring (FCs) differently modulate lipid metabolism and adipogenesis in 3T3-L1 adipocytes. The ^1^H-NMR profile of the extracts showed that FCs have a higher content of caffeic acid derivatives, glucose, and sucrose than FCa. In contrast, FCa showed a higher concentration of malic acid and furanocoumarins, identified as psoralen and bergapten. In vitro testing showed that only FCa treatments were able to significantly decrease the lipid content (Ctrl vs. FCa 25 μg/mL, 50 μg/mL and 80 μg/mL; *p* < 0.05, *p* < 0.01 and *p* < 0.001, respectively). Furthermore, FCa treatments were able to downregulate the transcriptional pathway of adipogenesis and insulin sensitivity in 3T3-L1 adipocytes. In more detail, FCa 80 μg/mL significantly decreased the gene expression of PPARγ (*p* < 0.05), C/EBPα (*p* < 0.05), Leptin (*p* < 0.0001), adiponectin (*p* < 0.05) and GLUT4 (*p* < 0.01). In conclusion, this study further supports an in-depth investigation of *F. carica* leaves extracts as a promising source of active compounds useful for targeting obesity and diabetes.

## 1. Introduction

Metabolic syndrome and diabetes pose a significant health burden worldwide and their incidence was found to be highly increased due to recent and alarming changes in food habits and lifestyle [1]. Metabolic syndrome is described as a pre-diabetic condition that includes obesity, dyslipidemia, impaired fasting glucose and/or impaired glucose tolerance, and reduced insulin sensitivity [2]. Diabetes mellitus (DM) is instead a chronic condition characterized by abnormal glucose homeostasis, resulting in elevated plasma glucose levels [3]. When prolonged in life, chronic hyperglycemia is associated with a myriad of long-term DM-related clinical sequelae such as damage to pancreatic β cells and microvascular complications, including retinopathy, nephropathy, and neuropathy [3].

The International Diabetes Federation has estimated that 463 million adults lived with diabetes worldwide in 2019, with a projected increase to 700 million by 2045 if effective prevention or strict glycemic control methods are not be adopted in DM patients before disease progression [4]. 

Several risk factors are known to be involved in the development of diabetes, including genetic factors (i.e., family history), lifestyle, diet, age, ethnicity, high blood glucose during pregnancy, hypertension, dyslipidemia, and obesity [5]. Of note, the increasing literature highlights a contributing role of adipose tissue dysfunction and hyperproliferation in inducing metabolic syndrome and diabetes [6]. 

Adipose tissue is highly dynamic and is the largest organ in humans involved in lipid storage and mobilization on the basis of energy requirements. It consists of several types of cells, including mature adipocytes that can increase in number and size. Decreasing proliferation and adipogenesis at the early stage of adipocyte differentiation might represent potential targets for preventing or treating obesity and, in turn, diabetes [7]. 

Besides insulin supplementation, to reduce the cluster of inter-related events of hyperglycemia and hyperlipidemia, a multi-targeted approach controlling both glucose and lipid metabolism is needed. The currently available non-pharmacological options are exercise or dietary modification [8], while pharmacological options include treatment with anti-diabetic, and anti-hyperlipidemic drugs [3]. 

Unfortunately, these latter therapeutic treatments have undesirable side effects, leading to a growing interest in traditional medicine to treat metabolic disorders, focusing mainly on herbal medicines [9,10]. Noteworthy is the development of metformin, one of the main anti-hyperglycemic agents, which can be traced back to the traditional use of *Galega officinalis* to treat diabetes [11,12]. With specific regard to the treatment of diabetes, obesity, and blood glucose control, several traditional plants with anti-diabetic and anti-obesity activities have already been reported. Among them, *Ficus carica* L. is considered to be one of the most promising [13,14]. 

*Ficus carica* is the most popular species of the genus *Ficus* (Moraceae family). It is native to Southwest Asia but has been widely cultivated throughout the Mediterranean basin since ancient times. Overall, the genus *Ficus* is one of the most abundant and varied genera of Angiosperms, including more than 800 species widely distributed in the tropical and subtropical belt of Southeast Asia and South America [15,16]. Traditional systems of medicine such as Ayurveda, Unani, and Siddha use different organs of multiple *Ficus* species such as *F. benghalensis*, *F. religiosa*, *F. glumosa*, *F. deltoidea*, *F. racemosa* and *F. carica* to treat metabolic, respiratory, gastrointestinal, reproductive and infectious disorders [17,18]. In support of traditional use, the bioactivity of *F. carica* extracts was also the object of several studies demonstrating its numerous pharmacological properties, among which include antioxidant, anticancer, anti-inflammatory and, remarkably, antidiabetic and hypolipidemic properties, in both in vitro and in vivo models [19,20]. More specifically, the data in the literature identify *F. carica* leaf extracts as an optimal source of anti-diabetic, anti-hyperglycemic and anti-obesity compounds that can be used as food supplements in diabetes prevention [16,21]. 

The biological activities of *F. carica* are mainly associated with the presence of heterogeneous phytoconstituents such as phytosterols, anthocyanins, amino acids, organic acids, fatty acids, phenolics, furanocoumarins hydrocarbons and aliphatic alcohols, which have been found in the latex, leaves, fruits and roots [22]. Nevertheless, contradictory results are present, probably due not only to the lack of standardization but also to the high variability in the phytoconstituents. Both these parameters can be influenced by diverse factors such as different preparations of the plant extracts, plant harvesting seasonality, drought stress and soil conditions [23,24,25].

Therefore, the main objective of this study was to evaluate whether the effects of *F. carica* leaf extracts on glucose and lipid metabolism in fully differentiated 3T3-L1 adipocytes depended on different phytochemical profiles that, in turn, were potentially due to different harvesting seasons. *Ficus carica* leaves were collected in spring and autumn (before and after fructification), the differences in metabolite content were explored by ^1^H-NMR profiling and the activity of the extracts as key adipogenic gene modulators with anti-adipogenic and anti-obesity properties was tested in an *in vitro* model. 

## 2. Materials and Methods

### 2.1. Plant Material and Extract Preparation

*Ficus carica* leaves were collected in November (FCa) and May (FCs) 2020 from a tree growing in the botanical garden of Bologna “Orto dei Semplici”. Voucher specimen (BOLO0602009) were retained in the Herbarium of Alma Mater Studiorum-University of Bologna. The leaves, deprived of the stalk, were dried in stove for three days at 40 °C and consequentially grinded. For ^1^H-NMR profiling, 30 mg of plant material were extracted with 1 mL of mixture (1:1) of phosphate buffer (90 mM; pH 6.0) in H_2_O-*d*_2_ (containing 0.1% TMSP) and MeOH-*d*_4_ by ultrasonication (TransSonic TP 690, Elma, Germany) for 20 min at 45 °C. After this procedure, samples were centrifuged for 10 min (17,000× *g*), then 700 μL of supernatant were transferred into NMR tubes. For bioactivity tests, 120 mg of plant material were extracted in 6 mL of MeOH:H_2_O (1:1), then sonicated for 30 min and centrifuged for 20 min. The supernatant was then divided into 3 × 2 mL test tubes and dried in a vacuum concentrator (Savant SpeedVac SPD210, Thermo Fisher Scientific, Waltham, MA, USA). Finally, it was resuspended in DMSO and then tested at different concentrations on cells.

### 2.2. Liquid-Liquid Partition of Ficus Carica Extracts

In order to elucidate the structure of the furanocoumarins visible in the ^1^H NMR profile of the extracts, FCa and FCs were subjected to a pre-purification procedure: 2 g of powder material were extracted in 100 mL of MeOH:H_2_O (1:1), then sonicated for 30 min and centrifuged for 15 min. The supernatant was dried in a rotary evaporator yielding 400 mg of extract, which was resuspended into 30 mL of water to undergo liquid-liquid partition with CHCl_3_ and EtOAc used in sequence, repeating the procedure three times for each solvent. The three fractions were dried in a rotary evaporator and analyzed by ^1^H NMR. The chloroform fraction contained the diagnostic signals of the furanocoumarins; therefore, it was subjected to 2D NMR analysis and the two major compounds were identified as psoralen and bergapten 

### 2.3. NMR Analysis

^1^H NMR spectra, J-resolved (J-res), ^1^H-^1^H homonuclear (COSY) and inverse detected ^1^H-^13^C correlation experiments (HMBC, HSQC) were recorded at 25 °C on a Varian Inova (Milan, Italy) instrument (equipped with a reverse triple resonance probe). For ^1^H NMR profiling the instrument was operated at a ^1^H NMR frequency of 600.13 MHz, and H_2_O-*d*_2_ was used as the internal lock. Each ^1^H NMR spectrum consisted of 256 scans (corresponding to 16 min) with the relaxation delay (RD) of 2 s, acquisition time 0.707 s, and spectral width of 9595.8 Hz (corresponding to δ 16.0). A pre-saturation sequence (PRESAT) was used to suppress the residual water signal at δ 4.83 (power = −6 dB, pre-saturation delay 2 s). The spectra were manually phased and baseline corrected and calibrated to the internal standard trimethyl silyl propionic acid sodium salt (TMSP) at δ 0.0 using Mestrenova software (Mestrelab Research, Spain). The analysis of ^1^H-NMR profiles of extracts was performed based on an in-house library and in comparison with the literature [26,27,28,29].

### 2.4. Cells Culture and Treatment

3T3-L1 murine pre-adipocyte cells, purchased from Sigma Aldrich (Sigma-Aldrich, St. Louis, MO, USA), were cultured in a Dulbecco’s Modified Eagle’s Medium (DMEM) containing 10% FBS (Sigma Aldrich) at 37 °C and 5% CO_2_ until ~80% cells confluence. Then, cells were seeded at a density of 1 × 10^4^ per well in 24-well plates. After 48 h, ~80% confluence was reached and cells were induced to differentiate (DAY 0) in MDI (Methylisobutylxanthine, Dexamethasone, Insulin) induction medium for 2 days. In particular, MDI induction medium is composed of 0.5 mM 3-isobutil-1-methylxantine, 1 μM dexamethasone and 10 μg/mL insulin in DMEM containing 10% FBS. At differentiation on DAY 3, the MDI medium was switched to a growth medium supplemented with 10 μg/mL insulin (DMEM containing 10% FBS and 10 μg/mL insulin). Medium was replaced every 2 days until fully differentiated adipocytes were obtained (DAY 11) [30,31]. Thus, at DAY 11, >90% of cells showed the characteristic adipocyte phenotype with accumulated lipid droplets. To test the effect of 48 h treatments with FC extracts, 3T3-L1 adipocytes were treated at DAY 9 with different dosages (25, 50, 80 and 100 μg/mL) of FCa and FCs and after 48 h (DAY 11), the MTT test was performed. 

### 2.5. Cell Viability

The 3T3-L1 preadipocytes were seeded at a density of 1 × 10^4^ cells per well in 96-well plates and incubated in culture medium. The cells were then treated with different dosages (25–100  μg/mL) of FCa and FCs. After 48 h, cells were incubated with 500 mg/mL of MTT (3-4,5-dimethylthiazol-2-yl)-2,5-diphenyltetrazolium bromide) for 1h at 37 °C. The supernatants were then removed and DMSO was added to each well. Plates were agitated to dissolve the formazan crystal products. The absorbance at 595 nm was measured using a Bio-Rad 3350 microplate reader (Bio Rad Laboratories, Richmond, CA, USA). The percentage of viable cells was calculated by defining the viability of untreated cells as 100%.

### 2.6. Red O Staining

3T3-L1 adipocytes were washed twice with PBS and fixed with 4% paraformaldehyde in PBS for 15 min at room temperature. After fixation, cells were washed two times with PBS and one time with 60% isopropanol. Then, cells were stained with filtered Oil-Red-O working solution (working solution: 60% Oil-Red-O stock solution and 40% distilled water) for 1 h at room temperature, washed 4 times with distilled water, and plates were dried and scanned for images. Oil-Red-O dye was then extracted using 100% isopropanol alcohol and measured spectrophotometrically at 490 nm.

### 2.7. Quantitative Real-Time PCR

Total RNA was extracted from 1 × 10^4^ cells following TRIzol^®^ reagent protocol (Invitrogen Corporation, Carlsbad, CA, USA). Then, 2μg of total RNA was retrotranscribed with M-MLV reverse transcriptase (Promega, Madison, Wisconsin, USA), using oligo-dT (15–18) as a primer in a final volume of 40μL reaction mix containing 1X RT buffer with MgCl_2_ 5 mM, DDT 10 mM, oligo-dT 5 mM, dNTPs 1 mM, RNase inhibitor 1U/mL and Reverse Transcriptase (M-MLV Reverse Transcriptase, Invitrogen) 10 U/mL. The reaction was incubated at 70 °C for 10 min and then at 4 °C for 2 min, followed by 37 °C for 60 min as reported in [32]. SYBR Green-based Real-Time PCR was used to determine cDNA levels. PPAR-γ, C/EBP-α, Adiponectin, Leptin, GLUT4, FAS and β-actin primers reported in Table 1 were provided by Metabion (Metabion International AG, Planegg, Germany). β-actin was used as the endogenous reference. Quantitative RT-PCR was performed with the ViiA7 Real-Time PCR System (Applied Biosystems, Foster City, CA, USA) using the iQ™SYBR Green Supermix method (Bio-Rad Laboratories, Richmond, CA, USA) according to the manufacturer’s instructions. Samples were run in triplicate in a 25 μL of reaction mix containing 12.5 μL x SYBR Green Master Mix (Bio-Rad), 12.5 pmol of each forward and reverse primer and 2 μL of diluted cDNA. The PCR mixtures were incubated at 95 °C for 10 min, followed by 40 cycles at 95 °C for 15 s and 60 °C for 60 s. A subsequent dissociation curve analysis verified the Ct for the target gene minus the mean of the C_t_ for the internal control gene. The Ct represented the mean difference between the Ct of the test minus the C_t_ of the calibrator. The N-fold differential expression in the target gene of the test compared with the calibrator (β-actin) was expressed as 2^−ΔΔCt^. Data analysis and graphics were performed using GraphPad Prism 9 software (GraphPad, San Diego, CA, USA).

### 2.8. Statistical Analysis

Statistical differences were determined by the analysis of variance (one-way ANOVA) followed, when significant, by an appropriate post hoc test; value of *p* ≤ 0.05 was considered statistically significant. The results are reported as mean ± standard error mean (SEM) of at least three independent experiments.

## 3. Results

### 3.1. NMR Profiling and Compounds Identification

This work represents the first report of ^1^H-NMR profiling of *Ficus carica* leaves. As reported in Figure 1, the main metabolites identified in the samples were both primary such as: sucrose, α-glucose, β-glucose, malic acid, alanine and valine, and secondary such as: bergapten, psoralen and a caffeic acid derivative.

The structure of the furanocoumarins (bergapten and psoralen) was confirmed by further 2D NMR experiments (Table 2 and Table 3 and Appendix A) performed on a pre-purified chloroform fraction obtained by liquid/liquid partition. The leaves collected in May (FCs) showed higher content of the caffeic acid derivative, glucose and sucrose compared to the sample collected in November (FCa). Conversely, the concentration of malic acid and furanocoumarins is higher in FCa than FCs. The quantity of psoralen was always greater than that of bergapten. 

### 3.2. In Vitro Evaluation of Cell Viability and Biocompatibility of FCa and FCs Treatments

From the phytochemical characterization of *Ficus carica* leaf extracts collected in autumn and spring, important differences in their composition were highlighted. Therefore, it was important to evaluate whether these variations in phytochemical composition consequently reflect different biological activities. Hence, at first, the possible effects of FC treatments on cell viability, biocompatibility and cell metabolic activity were assessed with MTT assay (Figure 2). MTT assay measures the activity of a mitochondrial enzyme and the colorimetric signal generated is an index of the number of viable cells and their metabolic activity [33].

Specifically, 3T3-L1 adipocytes were treated at different concentrations of FCa ranging from 25 μg/mL to 100 μg/mL for 48 h. MTT assay results showed that FCa treatment does not induce any cytotoxic effect on 3T3L-1 up to a concentration of 80 μg/mL, whereas dosages higher than 80 µg/mL resulted in cell suffering with a statistically significant decrease in cell viability (*p* < 0.05) (Figure 2A). On the contrary, FCs treatment showed good cell viability for all the concentrations tested (25, 50, 80 and 100 μg/mL) (Figure 2B) and interestingly, the lowest FCs dosage used (25 μg/mL) significantly increased cell metabolic activity (*p* < 0.05). Therefore, FC dosages at which no toxicity and good biocompatibility and cell viability was observed (25, 50 and 80 μg/mL) were selected for the following experiments. 

### 3.3. FCa and FCs Differently Modulate Lipid Accumulation in Mature Adipocytes

To further investigate the potential of *Ficus carica* phytoextracts in modulating lipid accumulation, 3T3-L1 preadipocyte were induced to differentiate by using MDI induction medium which contained pro-adipogenic factors such as insulin, dexamethasone, and isobutylmethyl xanthine. Different concentrations of FCa and FCs (25, 50, 80 μg/mL) were added to the medium at DAY 9 to observe the effects of 48 h treatments on mature adipocytes formation and lipid droplets accumulation. Thus, adipocytes treated and untreated with FC phytoextracts were stained by Oil Red-O at DAY 11. 

Figure 3 shows that FCa treatments, but not FCs, were able to induce a dose-dependent decreasing trend of adipocytes stained with Oil Red-O that specifically binds to fatty acids and lipid depots [34]. In order to semi-quantitatively assess lipid accumulation in cultured adipocytes, Oil Red-O dye was extracted with isopropanol and then evaluated spectrophotometrically. As shown in Figure 3A, FCa treatment was able to significantly reduce lipid accumulation by 20% at the lowest dosages (25 μg/mL; *p* < 0.05), up to a reduction of nearly 40% at the highest dosage used (80 μg/mL; *p* < 0.001). On the contrary, no significant differences were observed for FCs treatment for any of the dosages used (Figure 3B). 

### 3.4. FCa and FCs Modulate Genes Involved in Adipogenesis and Adipocytes Maturation

Considering the results obtained from cell cytotoxicity and lipid accumulation, FC phytoextracts treatments at both 50 and 80 μg/mL were chosen to further investigate whether FCa and FCs were able to modulate genes involved in adipogenesis, adipocyte differentiation and lipid accumulation. In this regard, the gene expression of key adipogenesis activators such as Peroxisome proliferator-activated receptor γ (PPARγ) and CCAAT/enhancer-binding protein α (C/EBPα) were investigated [35]. They mutually induce the expression of each other and have been reported to cooperate in the activation of a few adipogenic marker genes as Leptin, Adiponectin and GLUT4 [36]. In particular, our results demonstrated that FCa significantly decreased the gene expression of PPARγ (*p* < 0.05, Figure 4A) at both dosages used.

Considering the gene expression of C/EBPα, only a reducing trend was observed for 50 μg/mL of FCa treatment but became significant when 80 μg/mL of FCa was used (*p <* 0.01, Figure 4A). Similarly, a decreasing trend in Adiponectin gene expression was found for the 50µg/mL dosage of FCa, which became significant when FCa 80 µg/mL was tested (*p* < 0.05). FCa treatment was also able to highly reduce the gene expression of Leptin and GLUT4 at both dosages used. In details, FCa at 50 μg/mL significantly reduced the gene expression of Leptin (*p* < 0.01) and GLUT4 (*p* < 0.05) and their reduction is even more evident at higher dosage (*p* < 0.0001 and *p* < 0.01, respectively). Interestingly, FCa phytoextract was unable for any of the concentrations used to modulate the gene expression of FAS, an adipogenic enzyme involved in the de novo lipogenesis. 

Surprisingly, FCs phytoextracts showed an opposite effect when compared to FCa (Figure 4B). In fact, cells treated with FCs at 50 μg/mL showed a significant increase in PPARγ gene expression, while for C/EBPα, an increasing trend was found. These FCs effects were even more evident with increasing dosage of treatment. FCs at 80 μg/mL was able to induce a significant increase in the gene expression of PPARγ (*p* < 0.05) and C/EBPα (*p* < 0.05) and of their related target genes as Leptin (*p* < 0.05), Adiponectin (*p* < 0.05) and GLUT 4 (*p* < 0.01). Finally, not even FC phytoextract collected in spring were able to modulate FAS gene expression. Overall, these data suggest FCa phytoextract can exert an anti-adipogenic effect while FCs have shown an opposite pro-adipogenic-like effect on adipocyte differentiation and maturation.

## 4. Discussion

Targeting adipose tissue represents a valuable therapeutic strategy in obesity treatment and for diabetes prevention [37]. In fact, even though many therapies already exist for the treatment of adipose tissue-related disorders, nowadays nutritional interventions are recommended as first-line treatments and, among them, botanicals are considered promising coadjutants [18,38]. Despite the contradictory findings found in the literature [39], several studies have reported that *Ficus carica* L. extracts are endowed with potential health benefits against obesity-associated metabolic disorders and diabetes, but their mechanism of action is still debated [14,19] and the compounds responsible for the bioactivities have not yet been identified. Moreover, the plant metabolome is strongly affected by a number of biotic and abiotic factors, including seasonal variation [27]. The fluctuation in plant metabolome can be consequently reflected in their biological activities. In case of *F. carica* leaves, variations in its metabolites across the seasons were already reported. In particular, Innocenti et al. analyzed samples harvested from June to August, finding an increase in total coumarins content in *F. carica* leaves harvested in August [40]. Conversely, Marrelli et al. found that samples collected in June were endowed with a higher concentration of psoralen and bergapten than the ones collected in September [41]. 

This work was not designed to generically monitor the seasonal variation in *F. carica* metabolome, in fact, only two time points were chosen, therefore it is not possible here to draw general conclusions on seasonal variations. However, the ^1^H-NMR analysis suggested that coumarin content in the autumn extracts is higher than in the spring, while in the spring extracts there is a higher content of a caffeic acid derivative. The two samples not only showed differences in terms of phytochemical profile but also in their biological activity. In fact, in this study, only *F. carica* leaves harvested in autumn (FCa) were able to reduce lipid accumulation inside mature adipocytes. Moreover, FCa extract was able to modulate the expression of many genes involved in adipocyte differentiation, lipid accumulation and metabolism, whereas, notably, this occurs with opposite effects for *F. carica* leaves harvested in spring (FCs). In particular, in this work it has been demonstrated that FCa decreased the gene expression of PPAR-γ, the master regulator of adipogenesis, while FCs increased its expression. PPAR-γ plays a very important role, not only in adipocytes’ differentiation but also in lipid and glucose homeostasis [42]. It is well documented that compounds able to reduce PPAR-γ expression have anti-obesity effects [43]. In mature adipocytes, the reduced gene expression of PPAR-γ may contribute to decreased lipid accumulation, as also confirmed by our results in Red O Staining.

FC extracts were also found able to modulate the gene expression of C/EBP-α, another key adipogenic gene that acts as a marker of adipocytes differentiation and remains elevated during cell life [42]. Interestingly, C/EBP-α has a role in promoting lipid accumulation, thus the inhibition of C/EBP-α can aid in reducing lipid content. Our data showed that C/EBP-α expression was oppositely and significantly modulated according to FC’s seasonality, but only at the highest concentration (80 μg/mL) of both treatments. In detail, after 48 h treatment, a reduction in C/EBP-α gene expression was observed when FCa was used, while a decreasing tendency can also be noticed at the lower dosage. Differently, when leaves were harvested in spring, C/EBP-α gene expression was found to increase after 48 h of treatments.

Adipose tissue is also endowed with secretory properties as a huge number of proteins and other factors with biological activities can be secreted from adipocytes such as circulating hormones, pro-inflammatory and anti-inflammatory proteins [44]. Among these secreted factors, adipokines were found to play a role in the development of a metabolic syndrome and in adipose tissue expansion [45].

Among adipokines, we explored the potentiality of FC extracts in Adiponectin and Leptin gene expression modulation. Regarding Adiponectin, FCa caused a significant reduction in its expression only when the 80 μg/mL dosage of FCa was used, while at lower dosage (50 μg/mL) just a decreasing trend was observed. On the other thand, Adiponectin gene expression was increased after treating cells with FC harvested in spring at the highest dosage used. Adiponectin is also a pro-adipogenic marker [46], thus its reduction suggests an anti-adipogenic effect, while an opposite pro-adipogenic like effect can be attributed to its increased expression. These data supported that FC-dependent modulation of Adiponectin is upstream mediated by PPAR-γ modulation. In fact, it is well reported in the literature that adiponectin gene expression is under the control of PPAR-γ and C/EBP-α [46,47]. Of note, the seasonal opposite adipogenic effect of FC was also confirmed in the Leptin and GLUT4 gene expression data. Leptin is one of the adipokines responsible for regulating energy intake and expenditure [48]. The most well-known effect of Leptin is to regulate body weight, the feeding behavior of the animal, and energy balance [49], but it also has fundamental roles in tissue remodeling, inflammation, and glucose and lipid homeostasis [50,51]. GLUT-4 is the glucose transporter that is responsible for glucose uptake and whose function is disturbed in some conditions such as insulin resistance and type 2 diabetes [52]. During adipocytes differentiation, the gene expression of GLUT-4 favors glucose uptake and, in turn, glucose metabolism [53]. Thus, pharmacological drugs or natural compounds inhibiting both Leptin and GLUT-4 are highly desired for the treatment of obesity and diabetes. Interestingly, *F. carica* extracts did not modulate fatty acid synthase (FAS) levels, either when harvested in spring or in autumn. FAS is involved in the de novo lipogenesis [54] and, interestingly, it is not under the control of C/EBP-α-PPAR-γ axis [55,56], corroborating the hypothesis that PPAR-γ expression is a specific target gene of FC.

Overall, according to our results, FCa showed effective anti-adipogenic activities, by reducing lipid accumulation when 3T3-L1 mature adipocytes were treated for 48 h with FCa. These effects can be related to the FC phytochemicals content as demonstrated by the NMR spectra. In fact, most characterizing components, such as furanocoumarins were present only in FCa, which showed anti-obesity potentiality, while their absence was associated with reduced effectiveness: an almost opposite effect was observed. Furthermore, FCs NMR spectrum also revealed a higher presence of sucrose and glucose, two well-known energy sources involved in increasing the expression of pro-adipogenic regulators and fat storage [53,57], suggesting a highly relevant contributing role in exerting FCs effects on 3T3-L1 cells.

The data obtained in this work showed the promising activity of *F. carica* harvested in autumn as a potential source of anti-obesity and anti-adipogenic compounds, pointing out that this activity is strongly dependent on its phytochemical profile. These data lay the foundation for future investigations to fully explain seasonal variation in F. carica leaves metabolome, and how it could be reflected in the modulation of adipogenesis. Further biological experiments could also be conducted in vivo to investigate the possible adverse effects due to furanocoumarins content. In fact, different in vitro and in vivo studies have also demonstrated that furanocoumarins possess positive biological activities such as anti-inflammatory and anti-oxidative activities and bone health promoting effects [58,59]. However, they can also cause undesirable effects due to interactions with certain medications or by inducing phototoxicity [60,61].

In conclusion, this study further supports an in-depth investigation of *F. carica* leaves extract as a promising source of active compounds useful for targeting obesity and diabetes. However, whether the compounds contribute individually or accumulatively to the traditionally believed “medicinal” attributes of the plant is still unclear and under research.

## Figures and Tables

**Figure 1 nutrients-14-02833-f001:**
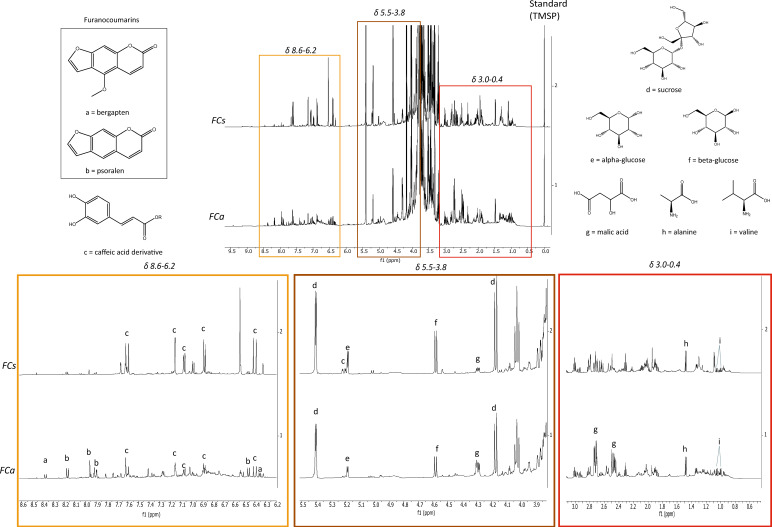
Comparison of ^1^H-NMR profiling of FCs and FCa: on top, entire spectrum; at the bottom, region from δ 8.6 to 6.2 (orange), region from δ 5.5 to 3.8 (brown), region from δ 3.0 to 0.4 (red) where a = bergapten, b = psoralen, c = caffeic acid derivative, d = sucrose, e = α-glucose, f = β-glucose, g = malic acid, h = alanine, i = valine.

**Figure 2 nutrients-14-02833-f002:**
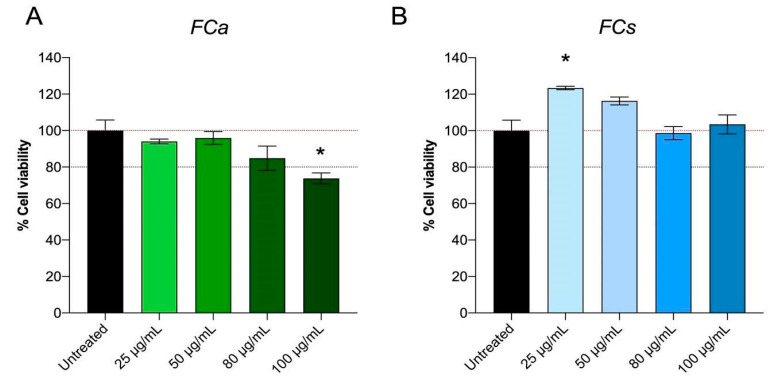
Effect of *Ficus carica* phytoextract collected (A) in autumn (FCa) and (B) in spring (FCs) on 3T3-L1 cell viability evaluated with MTT test. Data are shown as mean ± SEM; * *p*< 0.05 vs. control group (untreated).

**Figure 3 nutrients-14-02833-f003:**
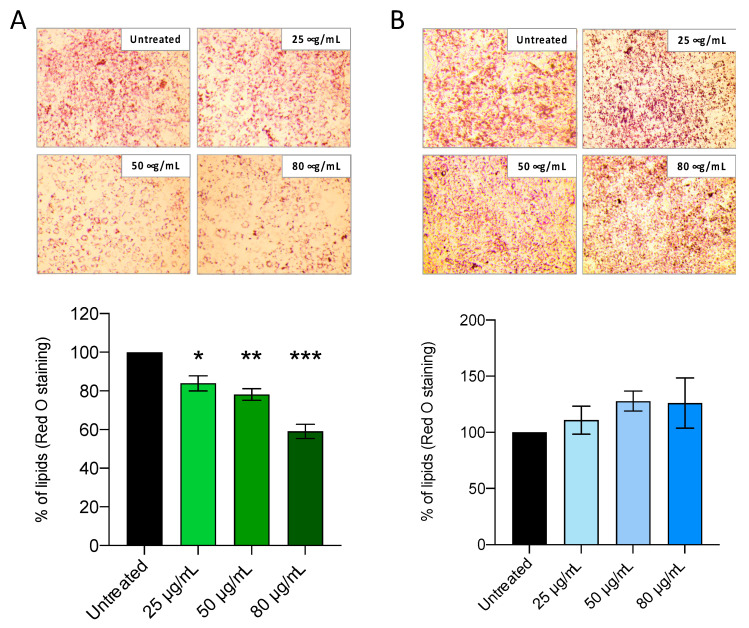
Effect of *Ficus carica* phytoextract collected (**A**) in autumn (FCa,) and (**B**) in spring (FCs) on 3T3-L1 lipid accumulation evaluated with Red O Staining. Data are shown as mean ± SEM; * *p*< 0.05; ** *p*< 0.01; *** *p*< 0.001 vs. control group (untreated).

**Figure 4 nutrients-14-02833-f004:**
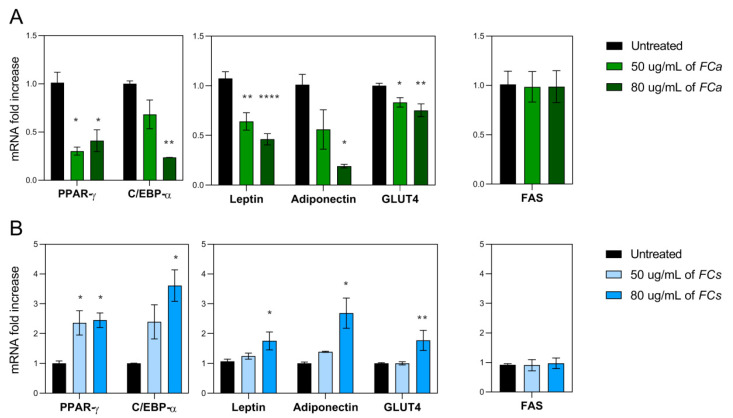
mRNA expressions of pro-adipogenic genes in 3T3-L1 adipocytes treated with *Ficus carica* phytoextract collected (**A**) in autumn (FCa,) and (**B**) in spring (FCs). mRNA expression of adipogenic transcription factors such as PPARγ and C/EBPα and of their related target genes as Leptin, Adiponectin and GLUT4. Additionally, the gene expression of the adipogenic enzyme FAS has been analyzed. Data are shown as mean ± SEM; * *p*< 0.05; ** *p*< 0.01; **** *p*< 0.0001 vs. control group (untreated).

**Table 1 nutrients-14-02833-t001:** Primers used for q-PCR.

Genes	Primer Sequences
Peroxisome Proliferator Activated Receptor Gamma (PPAR-γ)	f-5′-TCG CTG ATG CAC TGC CTA TG -3′; r-5′- GAG AGG TCC ACA GAG CTG ATT-3′
CCAAT Enhancer Binding Protein Alpha (C/EBP-α)	f-5′-GTA ACC TTG TGC CTT GGA TAC T-3′; r-5′-GGA AGC AGG AAT CCT CCA AAT A-3′
Leptin	f-5′-TCT TTC CGG AAC ATT TGG AG-3′; r-5′-TGT GAG ATC AAC CCT GGA CA-3′
Adiponectin	f-5′-GAA GCC GCT TAT GTG TAT CGC-3′; r-5′-GAA TGG GTA CAT TGG GAA CAG T-3′
Glucose Transporter type 4 (GLUT4)	f-5‘-GAT TCT GCT GCC CTG TC-3′; r-5′-ATT GGA CGC TCT CTC TCC AA-3′
Fatty Acid Synthase (FAS)	f-5′-AGA CCC GAA CTC CAA GTT ATT C-3′; r-5′-GCA GCT CCT TGT ATA CTT CTC C-3′
Actin (β-actin)	f-5′-AGC CAT GTA CGT AGC CAT CC-3′ r-5′-CTC TCA GCT GTG GTG GTG AA-3′

**Table 2 nutrients-14-02833-t002:** NMR spectral references of psoralen.

Position	Integrated Protons	^1^H, δ, m, J (Hz)	^13^C HSQC	HMBC Correlations	COSY Correlations
2	C	-	161.59	-	-
3	CH	6.37, d, *J* = 9.77	113.64	2,10	4
4	CH	8.03, d, *J* = 9.77	145.06	2,5,9	3,8,5
5	CH	7.88, s	120.28	4,7,8,9,3′	4,8
6	C	-	125.26	-	-
7	C	-	156.46	-	-
8	CH	7.52, s	98.92	6,7,9,10	3′,4,2
9	C	-	151.94	-	-
10	C	-	115.48	-	-
2′	CH	7.86, d, *J* = 2.33	147.23	6,3′	3′
3′	CH	6.95, dd, *J*1 = 2.33; *J*2 = 1.52	106.08	6,7,2′	8,2′

**Table 3 nutrients-14-02833-t003:** NMR spectral references of bergapten.

Position	Integrated Protons	^1^H, δ, m, J (Hz)	^13^C HSQC	HMBC Correlations	COSY Correlations
2	C	-	161.81	-	-
3	CH	6.25, d, *J* = 9.77	111.52	10,2	4
4	CH	8.26, dd, *J*1 = 9.77; *J*2 = 1.27	139.84	2,9	3,8
5	C	-	149.82	-	-
6	C	-	112.75	-	-
7	C	-	158.62	-	-
8	CH	7.15, s	92.76	6,7,9	
9	C	-	152.47	-	-
10	C	-	105.94	-	-
11	CH3	4.30, s	59.43	5	-
2′	CH	7.77, d, *J* = 2.29	145.17	6,7	3′
3′	CH	7.24, dd, *J*1 = 2.29; *J*2 = 1.27	104.89	6,7,2′	2′

## Data Availability

Data are available upon request to the corresponding author.

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
