# Peer review of "Different Seasonal Collections of Ficus carica L. Leaves Diversely Modulate Lipid Metabolism and Adipogenesis in 3T3-L1 Adipocytes"

_nutrients, 2022, doi:10.3390/nu14142833_

Round 1

Reviewer 1 Report

Dear authors,

Thanks for allowing us to review this manuscript by Pucci et al., entitled "Different seasonal collections of Ficus carica L. leaves diversely modulates lipid metabolism and adipogenesis in 3T3-L1 adipocytes." This work explored whether the harvest season of Ficus carica L. leaves affects their bio-activity of lipid synthesis inhibition in adipocytes in vitro. Extracts from plant leaves were subjected to NMR analysis to identify putative active ingredients. Cell culture studies were conducted to compare cell viability and adipogenic activities between FCa and FCs extracts, using MTT assays, histopathlogical staining, and RNAseq.

This work appears reasonably novel since there seem yet to be literature investigating the subject of treatment efficacy as a function of harvest season, using this particular combination of plant extract and cell line. Overall, we find the Methods, Results and Conclusions generally sound. This work could be of interest to the audiences in the fields of dietary supplements and herbal medicine.

Due to our area of expertise in bioNMR, we would focus our review on this aspect. The other reviewers are more suited to comment on plant and cell biology.

We only have a couple of questions/comments as follows.
1. Line 15: The current form of the Abstract section was largely qualitative (e.g. higher, lower). Based on the data, there were quantitative findings (e.g. the percentage changes of cell viability, lipid content and gene expression) that carry statistical significance. While it's ultimately up to the authors, we'd suggest that at least the key quantitative findings and associated numbers & p-values should be summarized in the abstract. Moreover, it would be good to report either the absolute or percentage difference between the experimental and control groups in the Results section, in addition to the p-values.

2. Line 105: Leaves from the same tree were harvested in two different seasons. Presumably the elapsed times from harvest to the in vitro experiments were different for the two samples. How were the leaves, or their extracts preserved during this period? Are the putative active ingredients chemically stable under the condition which the samples were preserved?

3. Line 105: Leaf sample was harvested from a single fig tree. Is there any evidence from the literature or data, that the findings from one tree is representative of all fig trees, i.e. the entire species?

4. Line 113: The techniques to extract leaf biosample was described. Is the described approach consistent with how it would be prepared, if intended as a dietary supplement?

5. Line 230: The qualitative difference of concentration was reported for putative bioactive ingredients between FCs and FCa. Wondering if it would be feasible to quantify these ingredients (e.g. using LCModel), and report either their absolute concentration, say in uM, or the relative differences by percent?

6. Table 2 & 3: While it is reasonable to report the 2D-NMR results in the table form, would it be possible to show the actual spectra if there's space? Or maybe in the supplemental data section?

7. Figure 1: The NMR spectra are small and therefore hard to read. Perhaps consider enlarge this figure, if there's space.

8. Line 266-270: The decrease of the MTT assay readings was interpreted as "decrease in cell viability", whereas its increase was interpreted as "higher cellular energy capacity". Wondering why this is so? In which way should the MTT assay be interpreted, viability or energy capacity? If the cell viability exceeds 100% (as in the right panel), does it mean the cells are proliferating?

9. Line 469-471: "These data confirmed that FC-dependent modulation of Adiponectin is upstream mediated by PPAR- γ modulation as adiponectin gene expression is under the control of PPAR-γ and C/EBP-α." The data showed the correlation between the two factors but not causation per se, or their upstream/downstream relations. Consider rephrasing this and/or cite references in which the upstream/downstream relationship is reported.

10. Figure 4: Interestingly, the FCs extract appeared to upregulate the adipogentic factors -- opposite to the intended effect. Wondering if there is a possible explanation for this? If the FCs have very little active ingredient -- as the NMR analysis in Figure 1 suggested, shouldn't it behave more or less like a placebo/control? Could there be other drivers of these gene modulations than the compounds identified in this work?

11. The following work seemingly explores a similar subject with a similar approach. Wondering if it could be beneficial to comment on whether there's a difference in putative mechanism of action or experimental methods between this work and the cited one, or was it just investigating difference species under the same genus?
Olaokun, O. O., McGaw, L. J., Awouafack, M. D., Eloff, J. N., & Naidoo, V. (2014). The potential role of GLUT4 transporters and insulin receptors in the hypoglycaemic activity of Ficus lutea acetone leaf extract. BMC complementary and alternative medicine, 14(1), 1-12.
https://link.springer.com/article/10.1186/1472-6882-14-269

12. Line 138: "1x104". Should this be "1x10^4"?
13. Line 192: " ...Table 1 and 2". Should this be "...Table 2 and 3"?
14. Line 433: "...is higher spring..." should be "...is higher than in spring..."
15. Line 498: "... further studies ... foresee also the inclusion of standard compounds found representatives ... bergapten." This sentence may need some grammatical attention - it seems hard to interpret.

Best regards,

Author Response

Dear authors,

Thanks for allowing us to review this manuscript by Pucci et al., entitled "Different seasonal collections of Ficus carica L. leaves diversely modulates lipid metabolism and adipogenesis in 3T3-L1 adipocytes." This work explored whether the harvest season of Ficus carica L. leaves affects their bio-activity of lipid synthesis inhibition in adipocytes in vitro. Extracts from plant leaves were subjected to NMR analysis to identify putative active ingredients. Cell culture studies were conducted to compare cell viability and adipogenic activities between FCa and FCs extracts, using MTT assays, histopathlogical staining, and RNAseq.

This work appears reasonably novel since there seem yet to be literature investigating the subject of treatment efficacy as a function of harvest season, using this particular combination of plant extract and cell line. Overall, we find the Methods, Results and Conclusions generally sound. This work could be of interest to the audiences in the fields of dietary supplements and herbal medicine.

Due to our area of expertise in bioNMR, we would focus our review on this aspect. The other reviewers are more suited to comment on plant and cell biology.

We only have a couple of questions/comments as follows.
1. Line 15: The current form of the Abstract section was largely qualitative (e.g. higher, lower). Based on the data, there were quantitative findings (e.g. the percentage changes of cell viability, lipid content and gene expression) that carry statistical significance. While it's ultimately up to the authors, we'd suggest that at least the key quantitative findings and associated numbers & p-values should be summarized in the abstract. Moreover, it would be good to report either the absolute or percentage difference between the experimental and control groups in the Results section, in addition to the p-values.

REPLY: We thank the reviewer for the very useful comments. The Abstract has been extensively revised and now the most important quantitative findings have been reported in the text.

2. Line 105: Leaves from the same tree were harvested in two different seasons. Presumably the elapsed times from harvest to the in vitro experiments were different for the two samples. How were the leaves, or their extracts preserved during this period? Are the putative active ingredients chemically stable under the condition which the samples were preserved?

REPLY: Samples were collected and dried in stove at 40°C, grounded and kept away from light and heat. When both samples were obtained they were simultaneously extracted and analyzed. We haven’t performed stability tests; however, we followed the general procedure used to treat and preserve medicinal plants. Generally, plant material is considered stable for at list one year if treated and preserved in this way and in our case the analyses were performed in a shorter time.

  1. Line 105: Leaf sample was harvested from a single fig tree. Is there any evidence from the literature or data, that the findings from one tree is representative of all fig trees, i.e. the entire species?

REPLY: As we specified in the manuscript our goal was not to draw general conclusion on fig tree and its metabolomic variation across the seasons. Moreover, besides the bibliographical references that we have already cited, there is no specific literature on this topic.

A more complex experimental designe would be necessary to answer the general question on fig metabolomic variation across the seasons (i.e. collect leaves from more than one tree and at more than two time points). It is our intention to develop this research, encouraged by our current findings. In this work we wanted to asses if Ficus carica was active in our test model and is the bioactivity  might be strongly variated sampling the leaves at two different time points. Once we have proved this we are now strongly motivated to deepen the investigation.

4. Line 113: The techniques to extract leaf biosample was described. Is the described approach consistent with how it would be prepared, if intended as a dietary supplement?

REPLY: In this work samples were extracted in MeOH:H2O, which is the most common solvent system used to analyze plant material. Of course, this procedure is not suitable to produce dietary supplements and further studies are always needed after the in vitro tests to understand how to prepare and administrate a botanical drug.

In this work we wanted to prove the bioactivity of fig leaves and we are aware that further studies should follow the current. For example MeOH might be replaced with EtOH, since EtOH:H2O is commonly used for herbal preparations and the extracts composition should not change that much. We cannot exclude the possibility to use the crude drug as it is, however further studies, including toxicological studies, are needed before defining these steps.

5. Line 230: The qualitative difference of concentration was reported for putative bioactive ingredients between FCs and FCa. Wondering if it would be feasible to quantify these ingredients (e.g. using LCModel), and report either their absolute concentration, say in uM, or the relative differences by percent?

REPLY: Of course, it is possible, and it would be possible to quantify them also by qNMR, defining the proper acquisition time. Our intention was to remain more general in this work and leave the absolute quantification of the metabolites of interest for a future work more focused on metabolomic variation of fig leaves.

  1. Table 2 & 3: While it is reasonable to report the 2D-NMR results in the table form, would it be possible to show the actual spectra if there's space? Or maybe in the supplemental data section?

REPLY: We  have now provided the 2D-NMR of the furanocoumarins in supporting material.

7. Figure 1: The NMR spectra are small and therefore hard to read. Perhaps consider enlarge this figure, if there's space.

REPLY: We  have now improved the quality of Fig. 1.

8. Line 266-270: The decrease of the MTT assay readings was interpreted as "decrease in cell viability", whereas its increase was interpreted as "higher cellular energy capacity". Wondering why this is so? In which way should the MTT assay be interpreted, viability or energy capacity? If the cell viability exceeds 100% (as in the right panel), does it mean the cells are proliferating?

REPLY: MTT assay was designed for proliferation and cytotoxicity assays, it measures the activity of a mitochondrial enzyme and the signal generated is dependent on the level of cell metabolism. Thus, the amount of signal generated is an index of the number of viable cells and their metabolic activity. Therefore, we agree with the reviewer that the phrase stated as it is in the text can mislead the reader, thus more information has now been included and the sentence now reports as follow:

“On the contrary, FCs treatment showed good cell viability for all the concentrations tested (25, 50, 80 and 100 μg/mL) (Figure 2B) and interestingly, the lowest FCs dosage used (25 μg/mL) significantly increased cell metabolic activity (p<0.05).

  1. Line 469-471: "These data confirmed that FC-dependent modulation of Adiponectin is upstream mediated by PPAR- γ modulation as adiponectin gene expression is under the control of PPAR-γ and C/EBP-α." The data showed the correlation between the two factors but not causation per se, or their upstream/downstream relations. Consider rephrasing this and/or cite references in which the upstream/downstream relationship is reported.

REPLY: A new citation has been included in the manuscript where it is better discussed that the adiponectin gene expression is mediated by PPAR-γ and C/EBP-α.

“These data supported that FC-dependent modulation of Adiponectin is upstream mediated by PPAR- γ modulation. In fact it is well reported in literature that adiponectin gene expression is under the control of PPAR-γ and C/EBP-α [42,43].

  1. Figure 4: Interestingly, the FCs extract appeared to upregulate the adipogentic factors -- opposite to the intended effect. Wondering if there is a possible explanation for this? If the FCs have very little active ingredient -- as the NMR analysis in Figure 1 suggested, shouldn't it behave more or less like a placebo/control? Could there be other drivers of these gene modulations than the compounds identified in this work?

REPLY: We thank reviewer for this comments. According with our results we cannot exclude that the regulation of adipogenic genes could be mediated by other bioactives besides bergapten and psoralen. In literature, additional downregultors of adipogenic factors can also be found in Ficus carica leaves such as angelicin, epiafzelechin or marmesin. Additional analytical analysis should be needed to better characterize these phytocostituents, of course they might be also metabolites at a very low concentration, thus not detected in the condition of our current analytical method.

Further, regarding the NMR analysis of FCs it is important to stress that it clearly reported a higher presence of sucrose and glucose that are two well-known energy sources involved in increasing the expression of pro-adipogenic regulators and fat storage [49,53],

  1. The following work seemingly explores a similar subject with a similar approach. Wondering if it could be beneficial to comment on whether there's a difference in putative mechanism of action or experimental methods between this work and the cited one, or was it just investigating difference species under the same genus?
    Olaokun, O. O., McGaw, L. J., Awouafack, M. D., Eloff, J. N., & Naidoo, V. (2014). The potential role of GLUT4 transporters and insulin receptors in the hypoglycaemic activity of Ficus lutea acetone leaf extract. BMC complementary and alternative medicine, 14(1), 1-12.
    https://link.springer.com/article/10.1186/1472-6882-14-269

REPLY: We thank reviewer for this suggestion. The suggested references have now been included in the text.

  1. Line 138: "1x104". Should this be "1x10^4"?
    13.Line 192: " ...Table 1 and 2". Should this be "...Table 2 and 3"?
    14. Line 433: "...is higher spring..." should be "...is higher than in spring..."
    15. Line 498: "... further studies ... foresee also the inclusion of standard compounds found representatives ... bergapten." This sentence may need some grammatical attention - it seems hard to interpret.

REPLY: We thank reviewer for their observations. All the requested modifications have been revised in the text.

Best regards,             

Reviewer 2 Report

Review report

Manuscript ID: nutrients-1769414

Different seasonal collections of Ficus carica L. leaves diversely modulates lipid metabolism and adipogenesis in 3T3-L1 adipocytes

The Authors conducted an in vitro study on 3T3-L1 adipocyte cell line demonstrating seasonal variation of Ficus carica L. leaves extract on lipid metabolism and adipogenesis. As was clearly demonstrated by the Authors, the profile of bioactive compounds from the plant is differing depending of time of harvesting (autumn versus spring) and demonstrate an opposite biological effect when analyzed the expression of pro-adipogenic genes expression. FCa plat extract reavealed an anti-adipogenic effect while FCs a pro-adipogenic-like effect. It seems to be dependent on the presence of furanocoumarins.

The study presents the high scientific level in terms of methods used, but some methodological aspects lower the value of obtained results.

My decision: major revision

The comments and suggestions are given below:

1.      In my opinion, research on plant material collected in one season is insufficient for the conclusions presented by the authors. I would fully agree with the statements of the conclusions, if the authors had performed the analysis with plant extracts obtained from at least two years in order to eliminate seasonal variability. In the presented study, the observed effects, while very promising, may be aleatory. Considering this, the authors should refer more strongly to the study limitation and should not express the conclusions with a such certainty.

2.      Please delete a dot from the title

3.      Line 79: please specify – what do you mean by “cytotoxic” in this sentence? A what type of the cells?

4.      Materials and methods: In my opinion, the order of subsection should be changed to be similar with the results: (1) Plant material and extract preparation; (2) Liquid-liquid partition of Ficus carica extracts; (3) NMR analysis; (4) Cells culture and treatment; (5) Cell viability test; (6) Red O staining; (7); Quantitative Real-Time PCR;

5.      Figure 4: the scale of Figures 4A and 4B should be similar (from 0 to 5) for better presentation of the results;

Author Response

Different seasonal collections of Ficus carica L. leaves diversely modulates lipid metabolism and adipogenesis in 3T3-L1 adipocytes

The Authors conducted an in vitro study on 3T3-L1 adipocyte cell line demonstrating seasonal variation of Ficus carica L. leaves extract on lipid metabolism and adipogenesis. As was clearly demonstrated by the Authors, the profile of bioactive compounds from the plant is differing depending of time of harvesting (autumn versus spring) and demonstrate an opposite biological effect when analyzed the expression of pro-adipogenic genes expression. FCa plat extract reavealed an anti-adipogenic effect while FCs a pro-adipogenic-like effect. It seems to be dependent on the presence of furanocoumarins.

The study presents the high scientific level in terms of methods used, but some methodological aspects lower the value of obtained results.

My decision: major revision

The comments and suggestions are given below:

  1. In my opinion, research on plant material collected in one season is insufficient for the conclusions presented by the authors. I would fully agree with the statements of the conclusions, if the authors had performed the analysis with plant extracts obtained from at least two years in order to eliminate seasonal variability. In the presented study, the observed effects, while very promising, may be aleatory. Considering this, the authors should refer more strongly to the study limitation and should not express the conclusions with a such certainty.

REPLY: We are aware of the limitation of this study and it was carried out to provide some preliminary results to encourage more in-depth analysis. Our intention, for the future research, is not to eliminate the impact of seasonality but to fully understand it in order to be able to provide the balsamic time for fig leaves in respect to our specific bioactivity. We have reformulated our conclusions, explaining the value of the preliminary results here obtained and what further studies should follow.

  1. Please delete a dot from the title

REPLY: We have correct this typo.

  1. Line 79: please specify – what do you mean by “cytotoxic” in this sentence? A what type of the cells?

REPLY: we really appreciate this punctual observation. In the phrase, “cytotoxic” is repetitive, as it is referred to the anti-cancer activity. Thus, “cytotoxic” has now been removed from the sentence.

  1. Materials and methods: In my opinion, the order of subsection should be changed to be similar with the results: (1) Plant material and extract preparation; (2) Liquid-liquid partition of Ficus carica extracts; (3) NMR analysis; (4) Cells culture and treatment; (5) Cell viability test; (6) Red O staining; (7); Quantitative Real-Time PCR;

REPLY: Now the method section has been re-ordered according to reviewer’s suggestion.

  1. Figure 4: the scale of Figures 4A and 4B should be similar (from 0 to 5) for better presentation of the results;

REPLY: We performed the corrections suggested from the reviewers and the new figures is here attached. Despite, we decided to keep the former figure as in this second version differences found in FCa are difficult to be appreciated.

(see file attached)

Otherwise, if the reviewer retains that the new figure improves the work, we will be glad to replace it.

Round 2

Reviewer 2 Report

The Authors properly addressed all my comments and suggestions. I recommend the manuscript for publication.